# Precision Immuno-Oncology in NSCLC through Gender Equity Lenses

**DOI:** 10.3390/cancers16071413

**Published:** 2024-04-04

**Authors:** Jennifer Marks, Arthi Sridhar, Angela Ai, Lauren Kiel, Rebekah Kaufman, Oyepeju Abioye, Courtney Mantz, Narjust Florez

**Affiliations:** 1Lombardi Comprehensive Cancer Center, Georgetown University, Washington, DC 20057, USA; jennifer.a.marks@medstar.net; 2Mayo Clinic, Rochester, MN 55905, USA; sridhar.arthi@mayo.edu; 3Olive View-UCLA Medical Center, University of California Los Angeles, Los Angeles, CA 90095, USA; ai.angela.c@gmail.com; 4Dana-Farber Cancer Institute, Boston, MA 02215, USA; lauren_kiel@dfci.harvard.edu (L.K.); rebekah_kaufman@dfci.harvard.edu (R.K.); oyepejuabioye@gmail.com (O.A.); courtney_mantz@dfci.harvard.edu (C.M.); 5Harvard Medical School, Boston, MA 02115, USA

**Keywords:** immunotherapy, precision oncology, clinical trials, female, women, underrepresentation, NSCLC

## Abstract

**Simple Summary:**

Precision immuno-oncology, the development of personalized treatments using the unique nature of an individual’s DNA, immune cells, and their tumor’s molecular profile, offers great promise. However, it has been well documented that biological sex considerably influences innate and adaptive immune responses, thereby conferring differences in the efficacy of lung cancer therapy between men and women. Despite this, women remain underrepresented in oncology clinical trials; this exclusion has resulted in our harboring of a limited understanding of the impact of immunotherapy on females and their survivorship, in addition to imprecise clinical recommendations. This review provides an overview of the sex-specific differences in immunity and immunotherapy efficacy, discusses contributing factors for the lack of women in clinical trials, and suggests future directions for precision oncology research, to ultimately aid in the development of treatment guidelines and recommendations that wholly consider the unique impact of immunotherapy on females and the intersectionality among them.

**Abstract:**

Precision immuno-oncology involves the development of personalized cancer treatments that are influenced by the unique nature of an individual’s DNA, immune cells, and their tumor’s molecular characterization. Biological sex influences immunity; females typically mount stronger innate and adaptive immune responses than males. Though more research is warranted, we continue to observe an enhanced benefit for females with lung cancer when treated with combination chemoimmunotherapy in contrast to the preferred approach of utilizing immunotherapy alone in men. Despite the observed sex differences in response to treatments, women remain underrepresented in oncology clinical trials, largely as a result of gender-biased misconceptions. Such exclusion has resulted in the development of less efficacious treatment guidelines and clinical recommendations and has created a knowledge gap in regard to immunotherapy-related survivorship issues such as fertility. To develop a more precise approach to care and overcome the exclusion of women from clinical trials, flexible trial schedules, multilingual communication strategies, financial, and transportation assistance for participants should be adopted. The impact of intersectionality and other determinants of health that affect the diagnosis, treatment, and outcomes in women must also be considered in order to develop a comprehensive understanding of the unique impact of immunotherapy in all women with lung cancer.

## 1. Introduction

Precision medicine involves tailoring interventions to patients by using their unique biomarkers within their genetic characteristics or molecular profiles [1]. Thus, in precision oncology, interventions extend beyond simply obtaining patient demographics and traditional clinical characteristics, and instead involve developing personalized treatment strategies using therapies targeted at a patient’s tumor’s molecular profile, as well as aiding in prognostication [2,3].

In tailoring interventions, it is important to consider sex disparities in the prognosis and outcomes of lung cancer, particularly non-small cell lung cancer (NSCLC). Compared to men, women are more frequently diagnosed with an adenocarcinoma subtype, at an earlier age, after less tobacco exposure, and are more susceptible to targeted approaches [4]. Women with NSCLC also show significantly better survival across all treatment modalities, regardless of stage or histology, though the magnitude of sex differences in mortality continues to decline due to increases in female mortality [4,5,6,7,8,9]. Nevertheless, it is crucial to recognize that cancer cells have the inherent capability to develop resistance to therapy, leading to a decreased duration of response and success of treatment. For instance, a wide variety of mechanisms, such as resistant EGFR-mutations, mutations in genes including PIK3CA, ALK, BRAF, KRAS, and TP53, and amplifications of MET and HER2 genes, are associated with acquired resistance to treatments such as osimertinib [10]. Understanding resistance mechanisms is an important aspect to improving treatment strategies.

Since the Human Genome Project, multiple large-scale studies have identified genomic, transcriptomic, and proteomic variations that may act as potential drug targets. We have also seen increased access to genomic sequencing on a patient-facing level, such that full genomic next-generation sequencing (NGS) can be completed in days and provide clinically relevant data [11]. With such advents of integrative genomics and advances in bioinformatics, we have been able to process an ample amount of genomic and transcription variations to identify potential targets with diagnostic and therapeutic implications [2,12]. This, combined with novel approaches to drug design, have contributed to the development of newfound therapies addressing targets previously thought to be “undruggable” due to factors such as their lacking defined ligand-binding pockets [13,14]. For instance, we may now target KRAS mutations due to the advent of selective KRASG12C inhibitors [13]. Additionally, there are multiple ongoing studies examining how to combine these selective inhibitors with other forms of therapy in addition to looking at non-G12C KRAS alleles [15]. Still, druggable driver mutations currently appear in low frequency amongst the cancer population, limiting the effectiveness of targeted treatment [16].

With advances in molecular immunology, immune checkpoint inhibitors (ICIs) have also become more sophisticated, with targeted immunotherapies more readily available. Even as immunotherapy becomes the first-line treatment for multiple types of cancer, though, there still remains a significant portion of patients who do not receive benefit [17,18], making it important to recognize immunotherapy-specific predictive biomarkers. One potential predictor of ICI response is the density of tumor-infiltrating lymphocytes (TILs) within a tumor [19,20,21] as well as natural-killer (NK) cells; in patients with advanced NSCLC receiving ICIs, lower NK cell values were independent prognostic factors for shorter OS and PFS. Sarcopenic patients with advanced NSCLC receiving ICIs also showed significantly worse PFS and had an 8.1 times higher risk of disease progression than non-sarcopenic patients [22,23]. As CXCR4 expression increases with stage progression in NSCLC, targeting the CXCL12/CXCR4 axis in immunotherapy has also emerged as a treatment approach, though this axis is not prognostic in early-stage NSCLC patients of either sex [24,25].

Another area garnering recent interest is a patient’s tumor microenvironment (TME) [26,27,28,29]. Notably, stromal cells are a cellular component within the TME and play a significant role in tumor metabolism, growth, metastasis, immune evasion, and treatment resistance [30]. There is emerging evidence that this stromal compartment can additionally shape antitumor immunity and responsiveness to immunotherapy [29,31]. For example, cancer-associated fibroblasts (CAFs) are the most common component of tumor stroma and are involved in tumorigenesis, tumor metastasis, and tumor angiogenesis [32,33,34]. One way that CAFs promote tumor progression is through the regulation of immune cells within the TME; strategies include inhibiting natural killer cells via the secretion of cytokines, chemokines, and MMPs [35]. Thus, several tailored strategies have been employed to combat these tactics, including anti-angiogenic therapy, immune modulation/reprogramming, CAF depletion, ECM targeting, and exosome or circulating tumor cell targeting, all of which are areas of active research [36].

Overall, immunotherapy has great promise, particularly considering how it is overall more effective and less toxic for patients [37,38,39,40]. However, the generalizability of current preliminary trial results are unclear due to their lack of representation of women [41,42]. Indeed, it is well documented that biological sex influences innate and adaptive immunity, as adult females typically mount stronger innate and adaptive immune responses compared to males [43,44,45,46]. Furthermore, for the majority of cancers, age-adjusted mortality rates and cancer-specific survival tend to be higher in males than females [47]. The implications of sex differences with regard to immunotherapy have not yet been well studied, offering opportunities for future research to provide more tailored data within populations that reflect our diverse clinical environment.

## 2. Sex Differences in Response to Immunotherapy

As previously noted, females typically express a more robust immune response, as they tend to have a higher B-cell response to various antigens and produce more interferons in macrophages and dendritic cells [48]. Sex hormones such as estrogen and testosterone also play a role in regulating the immune system; estrogen is thought to recruit immune cells that suppress the immune response, while androgens contrastingly may enhance it by promoting T-cell proliferation [48,49]. Estrogen-mediated immunomodulation also affects antigen-presenting cells and regulator T cells, with estrogen-enhancing PD-L1 expression [50]. Studies have further shown that sex chromosomes are determinants of sex dimorphism of anticancer immunity, owing to greater than fifty X-linked genes that play key roles in the innate immunity encoding for pro-inflammatory cytokines (e.g., TLR7 and TLR8) and in the regulation of adaptive immunity (e.g., IL2RG and IL13RA2) [51]. These X-linked genes are responsible for several transcription factors such as FOX-3 that are crucial for the development of regulatory T-cells and their functioning is responsible for suppressing immune responses [52]. Hormonal variations in women, such as those occurring during different phases of the menstrual cycle or menopause, may also influence immune responses and treatment efficacy [9,53]. Evidence suggests that females with early-stage NSCLC, specifically, also mount a stronger immune response due to their “hot” TME enriched with dendritic cells, CD4+ T cells, B cells, and a higher clonality of TILs. On the contrary, females with advanced NSCLC or those who are not treatment naïve have cancerous cells associated with complex resistance mechanisms and T-cell exhaustion, leading to the expression of multiple immune checkpoints; importantly, such differences are observed regardless of age, stage of disease, tumor histotype, and smoking status [54]. Sex differences in gut microbial composition may also play a role in the efficacy of immunotherapy. One study found that anti-PD-L1 could decrease the relative abundance of Lachnospiraceae, a bacteria associated with favorable responses to ICIs in female mice, while exerting no effect on male mice [55]. A higher relative abundance of Lachnospira has also been found in pre-menopausal women compared to post-menopausal women, who had similar levels to men [56]; the decline in microbiota diversity associated with aging might negatively influence ICIs [57]. Similarly, older adults have increased levels of regulatory T cells and pro-inflammatory cytokines such as interleukin (IL)-6, key mediators of immune evasion and resistance to ICIs. Though this decline might in principle result in an altered efficacy of immunotherapy in older patients, more research is needed; limited, non-generalizable subgroup analyses indicate that older patients may in fact gain the same benefit from immunotherapy as younger patients [57]. The breadth of recognized differences in the immunity between females and males has led to significant research aiming to determine if there is a difference in survival between women and men who are treated with ICIs. Several meta-analyses have been conducted with varying results.

A meta-analysis carried out by Conforti et al. explored sex-based differences in response to immunotherapy for lung cancer [58]. A prior meta-analysis conducted by this group included 20 randomized controlled trials (RCTs) and was the first to demonstrate a difference in the efficacy of ICIs in men and women [overall survival hazard ratio (OS-HR) 0.86 (95% CI 0.79–0.93) vs. 0.72 (95% CI 0.65–0.79) for females and males, respectively] [53]. In this subsequent study aiming to further investigate these findings, researchers integrated data from two meta-analyses, the first demonstrating how females derived a significantly greater benefit compared to males when receiving anti–PD-1/PD-L1 plus chemotherapy vs. chemotherapy for treatment across various solid tumors [0.48 (95% CI, 0.35–0.67) vs. 0.76 (95% CI, 0.66–0.87) for females and males, respectively]. The second meta-analysis included lung cancer trials alone and confirmed the prior observation that immunotherapy alone yielded greater benefits for males compared to females [0.78 (95% CI, 0.60 to 1.00) vs. 0.97 (95% CI, 0.79 to 1.19) for males and females, respectively]. Importantly, it also showed that the combined chemoimmunotherapy strategy proves more efficacious for females [0.44 (95% CI, 0.25–0.76)] compared to males [0.76 (95% CI, 0.64–0.91)], regardless of age, smoking status, or histology [58]. Kindred findings supporting the benefits of combined chemoimmunotherapy for females have been published in a meta-analysis conducted by Liang et al., which included 16 RCTs involving 10,155 patients with advanced NSCLC. This study demonstrated a more favorable OS-HR for women who were treated with ICI+ chemotherapy vs. chemotherapy alone when compared to males [0.63 (95% CI 0.42–0.92) vs. 0.79 (95% CI 0.70–0.89) for females and males, respectively] [59]. A later study conducted by Wu et al. encompassed 11 clinical trials (4 of which trials were for NSCLC) and corroborated the findings of Conforti et. al., demonstrating that males treated with ICIs were associated with a higher PFS and OS when compared to females, but notably, this difference was not observed with the NSCLC cohort [60]. Yet another meta-analysis that included only five phase-3 NSCLC trials (KEYNOTE 010, KEYNOTE 024, CHECKMATE 017, CHECKMATE 026, and CHECKMATE 057) comparing anti-PD1 ICI (pembrolizumab or nivolumab) to chemotherapy showed that a significant OS benefit was observed in males but not in females; this analysis, however, was limited by significant heterogeneity between studies and various cut-offs for biomarker expressions [61,62,63]. For instance, the inclusion criteria for the KEYNOTE 010, KEYNOTE 024, and CHECKMATE 026 trials involved PD-L1 tumor-expression positivity, while by contrast, the CHECKMATE 017 and CHECKMATE 057 trials enrolled patients with NSCLC without considering their PD-L1 status [64].

Despite the largely homogenous nature of the aforementioned data, several other meta-analyses examining the differences in the efficacy of ICIs among females and males have produced conflicting results [65,66]. For instance, a large-scale meta-analysis conducted by Wallis et al. encompassed 23 RCTs of patients, most of whom were in their 70s, and showed no statistically significant differences between the sexes when comparing ICIs to chemotherapy [65]. Boticelli et al. selected 36 clinical trials in which ICIs (anti-CTLA-4/PD-1/PD-L1) were studied and also did not appreciate any statistical differences in OS or PFS between the sexes; notably, however, this research did not include studies evaluating anti-PD-L1 ICIs in their final analysis [66]. Several other published studies (Table 1) have compared sex disparities in the outcomes for patients treated with ICIs. Despite varying results, a consistent theme noted continues to be the benefit for females when treated with combination chemoimmunotherapy; this contrasts with studies demonstrating that men experience a statistically significant benefit from ICIs alone compared to chemotherapy. It is important to recognize that limitations within all studies include the possibility for residual confounding factors such as the diverse range of tumor types considered, the absence of records pertaining to hormonal and PD-L1 status based on sex, and variations in biomarker expression cut-offs, which may influence the heterogenous results. Additionally, as meta-analyses rely on aggregate data instead of individual data, they may suffer from ecological fallacy. The inclusion of studies that are underpowered to explore the effect of sex disparities on outcomes, as well as the potential confounding impact of age on results, poses challenges, and more conclusive data are needed to delineate sex differences in response to ICIs. This includes addressing sex variations at the grassroot level and improving the gender-based diversity in the recruitment of patients to clinical trials.

## 3. Lack of Inclusion in Clinical Trials

Clinical trials are pivotal for shaping treatment protocols, yet persistent sex disparities exist affecting the inclusion of women. Indeed, research reveals a historical underrepresentation, with only 34.7% of participants in cancer preventive and therapy studies from 1990 to 2001 being female [69]. Though a more recent analysis involving data from 2008 to 2020 showed that females constituted 46.9% of participants in oncologic clinical trials, there were concerning proportional participation rates (PPR) of 0.912 across all trials, with women facing significant underrepresentation in surgical (PPR 0.74) and other invasive (PPR 0.69) oncology trials [70]. Further, recent immunotherapy trials such as the AEGEAN and KEYNOTE671, investigating perioperative chemoimmunotherapy in NSCLC, revealed a stark disparity of approximately 70% of participants in both treatment and placebo arms being male [71,72]. While the 2018 ADAURA trial did recruit over twice as many females as males (207 men, 471 women), it is important to note that this trial evaluated drug efficacy for an EGFR mutation, occurring over twice as often in females than in males (59% vs. 26%) [73]. Thus, appropriate representation is still needed for women when the mutation is not as prominent as EGFR. Underrepresentation also continues to be particularly pronounced among older women (>65 years) and those of minority race/ethnicity, though this trend extends to men as well [9].

The exclusion of women from clinical trials stems from a complex interplay of medical and societal considerations. A prominent factor is the potential impact of experimental treatments on fetal development, raising concerns related to pregnancy [74]. As women exhibit a higher incidence of autoimmunity and hypersensitivity reactions, caution is prompted regarding pregnant women’s inclusion in trials involving novel drugs or therapies [75]. Historical barriers also include a perception in the United States that women needed protection from clinical research, rooted in the adverse effects of thalidomide and diethylstilbestrol [69,74]. Past injustices such as the development of the HeLa cell line have also led to distrust of medical research [74]. Gender-specific barriers, including heightened time, financial costs, and increased familial responsibilities borne by women exacerbate such challenges [9]. The continuous lack of emphasis on women’s health research, limited awareness of trial opportunities, and fears related to randomization further compound the issue. More so, lingering gender-biased misconceptions that women are more difficult to recruit, are less willing to participate in trials, and are more difficult to work with [69] also persist as formidable obstacles to equitable representation [9,69]. For minoritized patient populations, additional barriers include stringent and narrow trial designs that may inadvertently exclude racial and ethnic minorities at a disproportionate rate [76,77], overly complex or lack of translated informational materials [78,79,80,81], and lack of local availability [76]. These patients may be hesitant to participate in clinical trials due to having a lack of trust in clinical research [82,83], as well as due to the financial toxicity [84,85] associated with trial participation, transportation needs, and other social determinants of health [86]. Owing to implicit biases, physicians also have less trust in and are less likely to offer a clinical trial to racial and ethnic minorities [87].

The repercussions of excluding women from clinical trials are extensive, impacting survivorship, treatment efficacy, sexual health, and fertility, and contribute to a lack of prospective validation for medication dosages and treatment protocols specifically tailored to women with lung cancer [9]. Consequently, women in targeted therapy trials are 25% more likely to experience severe adverse events than men, along with a heightened risk of underreporting [9,88,89,90]. Sex-specific variations in chemotherapy outcomes also reveal higher response rates but increased toxicities in women [91]. Immunotherapy, as previously noted, presents a different scenario, with men often experiencing greater efficacy, risk reduction, and benefits from the treatment alone, compared to women [53].

More so, limited prospective data assessing sexual dysfunction with cancer treatment, which remains underdiagnosed and undertreated [92], are available, resulting in a narrow understanding of how different cancer therapies affect libido, fertility, and overall sexual satisfaction in female patients [9]. Per the recently published SHAWL study, the largest study to date evaluating sexual health in patients with lung cancer, a resounding 77% of women experienced moderate to severe sexual dysfunction, with marked differences before and after cancer treatment [93]. However, these sexual health needs of women with cancer are often overlooked in clinical settings due to providers’ implicit discomfort, inexperience, or inaccurate perceptions of patient priorities [9,94]. In fact, a recent systematic study discovered that males with cancer had higher rates of patient–provider discussions of sexual issues than women [94]. Such disparities hinder the development of effective strategies to address issues such as vaginal dryness, pain during intercourse, or changes in body image [95].

Prospective information regarding the risks and long-term effects of cancer treatments on fertility is also lacking [96]. Compounded with the lack of general treatment guidance specifically tailored to women, this knowledge gap may result in even more poorly informed decision-making for women of reproductive age facing cancer diagnoses. Often, women may be unaware of fertility preservation options available to them or the impact of treatments on their ability to conceive in the future. Unsurprisingly, clinical recommendations about suitable treatments for pregnant patients with lung cancer are far from standardized, as most knowledge regarding treatment effects has been derived from research conducted in animal models or from data extrapolated from patients with breast cancer [97]. Though understudied, alkylating agents commonly used in lung cancer treatments have been associated with the highest risk of ovarian failure among cytotoxic chemotherapeutic medications [98], while the results from immunotherapy include the risk for hypogonadism, hypophysitis, hypothyroidism, low birth weight, increased miscarriage rates, stillbirth, and premature delivery [98,99]. Early-generation tyrosine kinase inhibitors, too, have been shown to negatively affect total follicle count, oocyte recovery, and ovarian reserve, reduce an embryo’s overall developmental potential, and even produce teratogenic effects [100]. Moreover, the evaluation of reproductive function and fertility is often insufficient when examining the consequences of cancer treatments [96]. As research reveals oncologists’ suboptimal knowledge, practices, and attitudes on fertility preservation and pregnancy during and after treatments [101,102,103], divergent survivorship experiences between men and women have been observed in discussions about fertility [9,96]. Unlike men, who were frequently advised to preserve sperm, women expressed negative sentiments regarding fertility preservation, citing inadequate information and presentation of their available options as contributing factors [9,104].

Despite the aforementioned implications, precision oncology discussions often overlook sex differences, which may unintentionally compromise the effectiveness of treatment strategies, limit informed decision-making, and impact the development of supportive care interventions (Figure 1). Without a diverse clinical trial participant pool, the efficacy, safety, pharmacokinetics, and pharmacodynamics of oncological treatments may not be adequately assessed for women, leading to disparities in outcomes [105,106]. The oversight of sex differences thereby undermines the fundamental principles of precision medicine, which aims to utilize treatments based on individual characteristics, including biological sex.

## 4. Future Directions

Sex-specific clinical research in women with lung cancer remains a pressing and unmet need (Figure 2). Future research initiatives should focus on sex-specific responses, hormonal influences, interactions with concurrent medications, and their effects on fertility, sexual health, survivorship, and overall outcomes [95,107]. These initiatives should also explore the interactions between immunotherapy and other medications commonly used by women, including hormone replacement therapies or contraceptives [108]. It is also imperative to scrutinize intersectionality; studying the diverse intersections of factors such as race, socio-economic status, and other determinants of health will shed light on disparities that affect the diagnosis, treatment, and outcomes in women with lung cancer [9,108].

A comprehensive approach involving healthcare providers, researchers, pharmaceutical companies, institutions, and governing bodies is also needed to promote a more unified understanding of clinical outcomes in women with lung cancer and enhance patient awareness and accessibility. Establishing standardized protocols and guidelines that consider sex-specific needs is necessary to ensure consistent and evidence-based care [96], but to increase the understandability of these guidelines, emphasis should be placed on community engagement and patient outreach. Implementing educational programs on cutting-edge therapies, treatment side effects, and sexual dysfunction to inform healthcare providers and women can empower patients to actively participate in treatment decisions [9]. Telemedicine initiatives can also enhance awareness and accessibility, particularly for those in remote or underserved areas, thereby enabling broader inclusion in clinical care and trials [9].

As socio-economic factors, access to care, and caregiver responsibilities can also affect women’s ability to enroll and remain engaged in clinical research, educating healthcare teams on the implications of clinical trial unenrollment is of paramount importance, as is adopting specific strategies for recruiting and retaining female participants. Measures such as flexible trial schedules, financial assistance, and transportation support can significantly enhance the participation rates among diverse groups of women [9,109]. Culturally sensitive and multilingual communication strategies should also be employed to reach a broader demographic, ensuring that research findings are representative of the diverse population affected by lung cancer. Involving patient advocates with clinical trial designs and utilizing patient navigators and community outreach may also help. Studies have also found that government-funded studies, as well as studies with first or senior female authors, had a higher percentage of female participants [9]. Furthermore, incentivizing clinical trialists and industry sponsors to address enrollment challenges could be implemented [109,110]. Although regulatory agencies have issued guidance to the pharmaceutical industry to study and reflect diversity within real-world populations, it is important to note that these directives are non-binding [110]. Innovative approaches, such as implementing hospital quality carrot-and-stick programs and drawing inspiration from industries outside of healthcare that have successfully tackled disparity gaps, can contribute to a more holistic and effective tactic in promoting equity [110].

## 5. Conclusions

The surging interest in precision immuno-oncology has highlighted a diverse range of promising therapeutic approaches for patients with cancer. Nevertheless, there remains a high priority to consider immunotherapy in the context of sex differences due to the more robust immune response of females, the immune-suppressing response of sex hormones such as estrogen, the vast array of X-linked genes that contribute to females’ immune-suppressing response, unique variations in the TME for females, and the observed sex differences in response to various lung cancer treatments. Though future large-scale, more standardized research is warranted to rectify conflicting results regarding the effectiveness of immunotherapy on females and males with lung cancer, we continue to observe an enhanced benefit for females when treated with combination chemoimmunotherapy in contrast to the more favorable approach of utilizing ICIs alone in men. Multi-disciplinary collaboration among healthcare providers, researchers, pharmaceutical companies, institutions, and governing bodies will be necessary to provide a more unified understanding of clinical outcomes in women with lung cancer.

Enhancing the effectiveness of treatment plans, guidelines, and recommendations that are specifically tailored to the unique needs of patients begins with improving the gender-based diversity of clinical trials. Sex-specific variations in immunotherapy, targeted therapy, and chemotherapy outcomes continue to be observed as a result of this inadequate representation; more so, treatment implications on issues related to sexual dysfunction, fertility, and other survivorship issues unique to women are largely unknown. To overcome persistent gender bias and societal obstacles facilitating the historic and current exclusion of women from clinical trials, efforts should be made to incorporate flexible trial schedules, multilingual communication strategies, and financial and transportation assistance for participants, as well as adopting strategies to incentivize trials to more equally represent the diverse body of patients with lung cancer. Of equal importance throughout all efforts is the need to consider the impact of intersectionality, including race, socio-economic status, and other determinants of health that affect the diagnosis, treatment, and outcomes in women with lung cancer. Such endeavors will aid in developing a more tailored understanding of the unique impact of immunotherapy on females, with an ultimate goal of constructing and implementing standardized protocols and guidelines that consider intersectionality, sex-specific needs, and opportunities across practices in various medical settings.

## Figures and Tables

**Figure 1 cancers-16-01413-f001:**
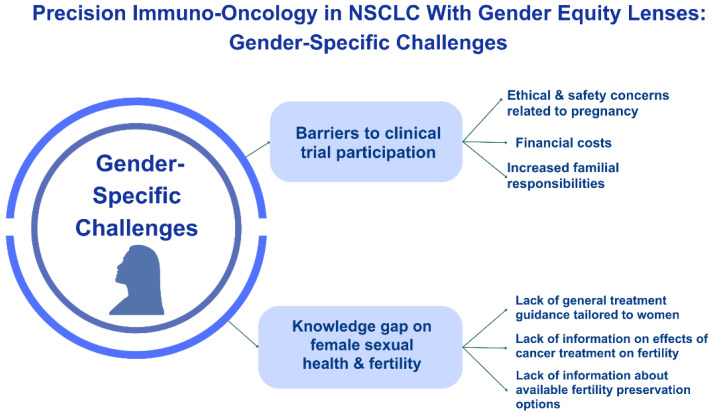
Gender-specific challenges of precision oncology are often based on the lack of clinical trial participation for women and our collective lack of knowledge regarding treatment effects on survivorship issues such as sexual health and fertility.

**Figure 2 cancers-16-01413-f002:**
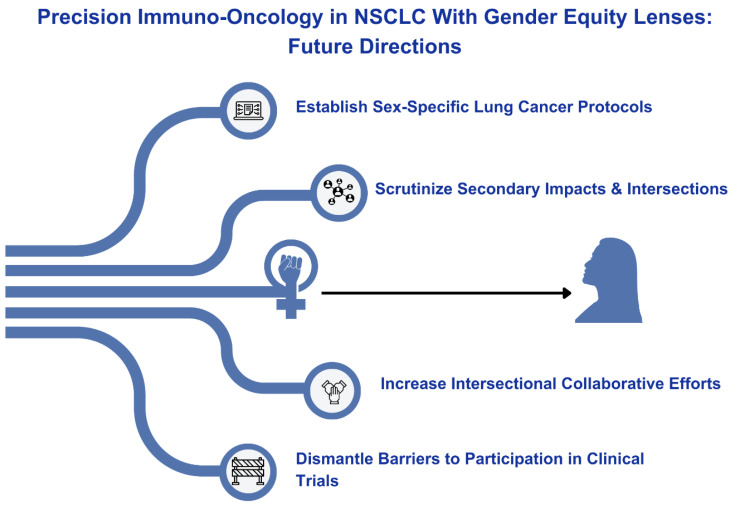
Future directions for precision immuno-oncology research.

**Table 1 cancers-16-01413-t001:** Studies evaluating the impact of gender on response to immune checkpoint inhibition.

Study	Type of Study	Sample Size	Cancers Included	Treatment Regimen	Outcome	Sex Differences in OS	Hazard Ratios (If Available)
[66]	Meta-Analysis	11 RCTs	Solid tumors	ICI vs. chemotherapy	OS, PFS	No significant difference in males and females noted for OS or PFS	Anti-PD1: OS (males vs. females: HR 0.72, 95% CI 0.64–0.83 vs. HR 0.81, 95% CI 0.70–0.94, *p* = 0.285) Anti PD-1 PFS: (males vs. females: HR 0.66, 95% CI 0.52–0.82 vs. HR 0.85, 95% CI 0.66–1.09, *p* =0.158).
[67]	Systematic Review & Meta-Analysis	21 RCTs, 26,598 patients	Solid tumors	ICI alone or with chemotherapy vs. chemotherapy	OS, PFS	Similar OS in males and females for anti-PD-1/PDL-1. Anti-CTLA-4 use was associated with longer OS in men only	OS: Females (HR, 0.77; 95% CI 0.67–0.89, *p* < 0.001) Males (HR, 0.73; 95% CI 0.66–0.80, *p* < 0.001)
[53]	Meta-Analysis	20 RCTs, 11,351 patients	Solid tumors	CTLA-4 or PD-1 inhibitors vs. chemotherapy	OS	Men experienced longer OS when compared to females	OS: Women (HR, 0.86; 95% CI 0.79–0.93), Male (HR, 0.72; 95% CI 0.65–0.79)
[58]	Meta-Analysis	8 RCTs, 574 NSCLC patients	NSCLC	PD-L1 or PD-L1 alone or with chemotherapy vs. chemotherapy	OS, PFS	Women had better OS with PD-1 and chemotherapy combination when compared to Males. Males had a better OS in the immunotherapy alone arm	OS PD-1/PD-L1 alone: Females (HR, 0.97; 95% CI = 0.79 to 1.19), Male (HR, 0.78 (95% CI = 0.60 to 1.00) OS combination: Females (HR, 0.44 95% CI = 0.25 to 0.76), Male (HR, 0.76; 95% CI = 0.64 to 0.91)
[65]	Meta-Analysis	23 RCTs, 13,271 patients	Solid Tumors	ICI vs. standard therapies	OS	Benefit noted in both men and women with no statistical difference noted between the sexes	OS: Females (HR, 0.77; 95% CI, 0.67–0.88; *p* = 0.002), Men (HR, 0.75; 95% CI, 0.69–0.81; *p* < 0.001)
[60]	Meta-Analysis	11 RCTs, 6096 patients	Solid tumors, (4 lung cancer RCTs)	CTLA-4 or PD-1 inhibitors vs. chemotherapy	OS, PFS	Better PFS and OS seen in males vs. females treated with ICI. However, this was not noted in the NSCLC cohort.	OS: Females (HR = 0.74; 95% CI, 0.65–0.84; *p* < 0.001) Males (HR = 0.62; 95% CI, 0.53–0.71, *p* < 0.001)
[59]	Meta-Analysis	16 RCTs, 10,155 patients	NSCLC	ICI alone or with chemotherapy vs. chemotherapy alone	OS	Those who received ICIs (with or without chemotherapy) had longer OS than those who did not receive ICIs and was comparable between both genders	Overall: Females (HR: 0.74, 95%Cl 0.63–0.87), Males (HR: 0.76, 95%Cl 0.71–0.81) ICI + Chemo: females (HR: 0.63, 95%Cl 0.42–0.92), males (HR: 0.79, 95%Cl 0.70–0.89) ICI alone: Females (HR: 0.83, 95%Cl 0.73–0.95), Males (HR: 0.74, 95%Cl 0.67–0.81)
[68]	Meta-Analysis	15 RCTs, 9583 patients	Lung cancer	ICI alone or with chemotherapy vs. chemotherapy alone	OS, PFS	Both females and males benefited from anti-PD-1 therapies and benefit was seen only for males with anti-PD-L1 therapies.	Anti-PD-1: Females (HR = 0.69, 95% CI, 0.52–0.93) Males (HR = 0.73, 95% CI, 0.67–0.80) Anti-PD-L1: Females (HR = 0.69, 95% CI, 0.44–1.07), Males: (HR = 0.80, 95% CI, 0.69–0.92)

## Data Availability

The data presented in this study are openly available via access to their corresponding citations found in the References section of this manuscript.

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
