# Peer review of "Precision Immuno-Oncology in NSCLC through Gender Equity Lenses"

_cancers, 2024, doi:10.3390/cancers16071413_

Round 1
Reviewer 1 Report
Comments and Suggestions for Authors
The article is consistent within itself. The authors describe the sex-specific differences in immunity and immunotherapy efficacy, discusses contributing factors for the lack of women in clinical trials, and suggests future directions for precision oncology research, to ultimately aid in the development of treatment. The references are relevant and recent. The cited sources are referenced correctly. Appropriate and key studies are included. The paper is comprehensive, the flow is logical and the data is presented critically.
However, there are some specific comments on weaknesses of the article and what could be improved:
1) The text reports that there are differences in the effectiveness of ICIs in men and women, please address this concept more broadly by reporting the differences found.
2) Several articles discuss gender differences in response to cancer treatments, disease progression and prognosis, please cite:
-Gender-related disparities in non-small cell lung cancer. (Marco G Paggi et al., 2010).
-Female gender is an independent prognostic factor in non-small-cell lung cancer: a meta-analysis. (Haruhiko Nakamura et al., 2011).
-Sex and gender differences in non-small cell lung cancer. (Donington JS, Colson YL, 2011).
3) Precision medicine involves tailoring interventions to patients by using their unique biomarkers within their genetic characteristics or molecular profiles, some studies have been conducted to identify specific predictive markers; please discuss the results reported in these papers:
-Circulating Natural Killer Cells as Prognostic Value for Non-Small-Cell Lung Cancer Patients Treated with Immune Checkpoint Inhibitors: Correlation with Sarcopenia (Tenuta M, et al., 2023).
-Impact of Sarcopenia and Inflammation on Patients with Advanced Non-Small Cell Lung Cancer (NCSCL) Treated with Immune Checkpoint Inhibitors (ICIs): A Prospective Study (Tenuta M, et al.,2021).
-CXCR4 expression in lung carcinogenesis: Evaluating gender-specific differences in survival outcomes based on CXCR4 expression in early-stage non-small cell lung cancer patients. (Andrea S Fug et L., 2021).
-Gender Differences in Long-Term Survival after Surgery for Non-Small Cell Lung Cancer (Yukihiro Yoshida et al., 2016).
Comments on the Quality of English Language
The authors should consider a proof-reading of their manuscript, as there are typos and English grammar mistakes.
Author Response
Please see the attachment of reviewer responses.

Reviewer 2 Report
Comments and Suggestions for Authors
This review examines evidence suggesting enhanced benefit for women with lung cancer treated with chemoimmunotherapy compared to immunotherapy alone favored in men. Contributing factors to female underrepresentation in trials are discussed, including ethical concerns and gender biases. The authors emphasize the need for greater inclusion of women to elucidate sex-specific effects, improve treatment strategies, and address unique supportive care needs. Increasing trial accessibility, community engagement, and considering intersectional factors are recommended to achieve precision, equitable cancer care. However, the manuscript requires some minor changes before being accepted for publication.
Specific comments:
1. Some sections could be more concise and focused, especially the introduction on precision oncology background.
2. Relatively limited discussion of biological mechanisms underlying observed sex differences beyond hormonal effects.
3. Line 76 - Define "undruggable" targets early on.
4. Lines 115-123 could be streamlined as an overview of sex differences in immunity.
5. Consider adding a limitations section to address study constraints.
6. Figure 1 is clear, but the caption could be more descriptive.
7. In general, the writing is clear but there are a few longwinded sentences that could be shortened for clarity.
8. Check for consistent use of abbreviations after defining them.
9. Ensure uniform formatting of references.
Author Response

(The authors gave the same response as above.)

Reviewer 3 Report
Comments and Suggestions for Authors
This review significantly contributes to the field of precision immuno-oncology by emphasizing the necessity of acknowledging sex-specific differences in immune responses and addressing the underrepresentation of women in clinical trials, particularly concerning NSCLC.
The recommendations and future directions proposed have the potential to enhance the development of more precise and unbiased cancer treatments.
Minor revisions may be warranted for clarity in some sections.
-The Introduction effectively outlines the aim of the review, covering key advancements in precision oncology and setting the stage for the discussion on sex-specific differences in immunotherapy efficacy. In addition to the advancements in all these therapies, it is crucial to recognize that cancer cells have the inherent capability to develop resistance mechanisms, including the acquisition of new mutations, as a response to treatment pressure. Understanding these mechanisms is also essential for improving treatment strategies. I believe that the authors should include this detail in the Introduction.
-I consider it important to also reference the age of the women reported in the studies, as age directly influences women's hormonal status (reproductive age, menopause, and postmenopausal period). Additionally, are there any data available from adolescent/young adult women?
-The authors have beautifully analyzed the underlying reasons why women are often excluded from trials. I believe there should be a mention regarding women belonging to minorities who are even less represented.
-Socio-economic factors, cultural beliefs, access to healthcare resources, and caregiver responsibilities can all affect women's ability to enroll and remain engaged in clinical research. Addressing these multifaceted barriers requires tailored recruitment strategies, patient education initiatives, and supportive policies that prioritize inclusivity and accessibility for women of all ages and hormonal statuses. Concerning future directions, the authors should report more extensively in the education and training of healthcare providers on the importance of including women in clinical trials. This includes raising awareness about sex-specific differences in disease presentation and treatment response, as well as specific strategies for recruiting and retaining female participants.
Author Response

(The authors gave the same response as above.)
